# Agro-morphological and genetic variability analysis in oat germplasms with special emphasis on food and feed

Raj Kumar[1☯], Smriti Varghese[1☯], Deepanshu Jayaswal [2]*, Kuldip Jayaswall[2]*, Kuldeep Yadav[1], Gaurav Mishra[1], R. P. Vyas[1], H. C. Singh[1], H. G. Prakash[1], Arvind Nath Singh[2], Sanjay Kumar[2]

1 Chandra Shekhar Azad University of Agriculture and Technology, Kanpur, India, 2 ICAR-Indian Institute of Seed Science, Mau, U.P., India

☯ These authors contributed equally to this work.
* jayaswaldeepanshu@gmail.com (DJ); kjgenome@gmail.com (KJ)

**Data Availability Statement:** All relevant data are within the paper and its Supporting Information files.

## Abstract

The gaining attention of underutilized oat crops for both food and feed, mining of quality and yield related genes/QTLs from available germplasms of oat is need of the hour. The large family of grasses has a vast number of germplasms that could be harnessed for bio-prospecting. The selection of cross-compatible oat germplasms by molecular markers could be used for the introgression of the novel traits into the elite background of oats. The process needs a thorough study of genetic diversity to see the evolutionary relatedness among germplasms. Considering this, in the present study, the genetic diversity of 38 oat germplasms with 12 agro-morphological traits was carried out using 22 Inter Simple Sequence Repeat (ISSR) markers. We found a high level of polymorphism and 158 distinctive alleles; on average 7.18 alleles per primer, further, high-yielding genotypes were identified with the help of phenotypic data and genetic diversity was analyzed by using DNA fingerprint-based principal component analysis, UPGMA dendrogram. Among these 38 germplasms; eight were identified as superior under high grain yield (OS-424, OS-403, NDO-1101, OL-10, UPO-212, OS-405, OS-6, and OS-346) and another eight germplasms were identified as superior for the high fresh weight (for fodder purpose, NDO-711, RO-19, OL-14, OL-1760/ OL-11, NDO-10, UPO-212, UPO-06-1, and RO-11-1). These results suggest that germplasms that are closely related (Cross-compatible) and have good potential for desirable traits could be used for varietal development by using marker-assisted selection.

## Introduction

Oats hold the sixth rank in the world cereal production statistics after wheat, maize, rice, barley, and sorghum. The common oat (*Avena sativa* L.) is a cereal cum fodder crop grown primarily for its green fodder and grains. It is rich in antioxidants such as α-tocotrienol, α- tocopherol, avenanthramides, and total dietary fiber including the soluble fiber β-glucan [1]. Oat has gained much attention owing to its higher nutritional value, excellent health functions, and gluten-free property [2]. Grain quality and yield with several grain characteristics are routinely used to define milling

**Funding:** The author(s) received no specific funding for this work.

**Competing interests:** The authors have declared that no competing interests exist.

quality [3]. Developing new oat varieties that combine high yield, β-glucan content and high great content is an obstacle, such that improvement in one trait tends to be accompanied by a reduction in others [4]. Oats face several biotic and abiotic constraints which drastically affect their fodder and grain yield. Serious diseases like crown rust [5], smut [6], stem rust [7], cold weather, low soil pH, and variation in nutrient and health conditions of the soil affect production of oats. Cultivars with an improved yield capability that can be also grown at higher elevations than the current climatic limits for arable agriculture could relieve some of the overgrazing pressures and destruction of forests. As the available land currently used for food production becomes scarce with the increasing population, and the need for greater food production increases, hence, more research is needed to develop special-purpose forage cultivars that fit specific end uses. There are few oat breeding programs where the primary objective is developing oats for forage, and very little work is underway to develop germplasm for conditions like the cool and high altitude regions found in the parts of the Himalayas. A point at issue is to consider the potential for quantum leaps in oat cultivar performance where the oat crop is a mandated food crop of an International Agriculture Research Centre, such as the International Center for Agricultural Research or the International Maize and Wheat Improvement Center. It would provide potential spin-offs by providing the basis for smaller oat breeding operators to develop germplasm specifically for grain, forage and fodder uses, including lowering the temperature threshold at which oat germplasm can grow, and producing higher yields of forage and fodder for livestock uses in cool regions [8]. Biochemical markers were used to study genetic diversity and relationships before the 1980s when molecular markers were unknown. Furthermore, the discovery of molecular markers has simplified the examination of genetic variation. Since molecular markers are more trustworthy and accurate than other marker systems, they are widely employed for genetic diversity analyses, phylogenetic investigations, and cultivar identification. Genetic polymorphisms in the cultivated *Avena sativa* L. are very low as a result of continuous selective breeding for crop improvement, therefore marker-assisted selection (MAS) and marker-assisted breeding (MAB) could play a key role in the production of new varieties. Genetic diversity study using multiple molecular markers in diverse crops has been done efficiently for many years [9–13]. Montilla et al., 2013 [14], had studied the vast number of oat accessions by using simple sequence repeat (SSR) and expressed sequence tag (EST) molecular markers to see the relatedness and difference at the allelic level and classified them based on genetic distance. Among other molecular markers, Inter Simple Sequence Repeat markers are widely employed for genetic diversity study because they have various advantages over other markers, including being PCR-based, requiring no sequence information, distributing across the entire genome, using a tiny amount of template DNA, and being cost-effective. ISSR marker genotypic data, pedigree analysis, and other phenotypic data can all help with germplasm selection for future crop development through molecular breeding initiatives [15]. ISSR markers are dominant in nature; hence, phenotypes can be differentiated at each locus in the Mendelian pattern.

In contemplation of achieving the adequate production of crops, it is important to spot the superior genotypes for both fodder and grain yield. To precise the research work and speedup up the oat improvement program, the genomic study will be helpful to get a reliable result. Hence to identify higher-yielding germplasms and to introgress superior yield governing genes to ameliorate the existing varieties the present investigation was carried out containing 38 oat germplasms.

## Material and methods

### Analysis of agronomic morphological and yield traits

The present study was carried out using 38 oat germplasms sourced from Indian Grassland and Fodder Research Institute Jhansi, Uttar Pradesh, India during the *Rabi* 2021 at Chandra

Shekhar Azad University of Agriculture and Technology, Kanpur, Uttar Pradesh, India (S1 Table). The experimental material consisted of 38 germplasms that were sown in Randomized Block Design with three replications in timely sown (TS) condition. The entries were sown in a single row plot of $(3 \times 1)$ m$^2$ area with inter and intra-row spacing of 30 cm and 5 cm, respectively. The observations were recorded on various quantitative characters *viz.* days to 50% flowering, plant height, tillers per plant, leaf per plant, leaf length, leaf width, fresh weight (for fodder purpose), leaf area index, dry weight, harvest index, and grain yield (for food purpose) (S2 Table). The emphasized traits of fresh weight and grain yield were observed at the climax stage of the plant. Three randomly selected plants in each row of each replication for all the characters were recorded under study. For all 38 genotypes, all character's quantitative values in three replications were taken and statistical analysis was performed (S2 Table) and the mean quantitative value was used for the development of a heat map (S1A and S2A Figs) and correlation heat map matrix using Mev software (S1B and S2B Figs) for grain and fodder yield respectively.

### DNA isolation

Fresh tender leaves of oats were collected from the agricultural field and DNA was isolated following the CTAB method of DNA isolation with minor modification [16]. The polyvinyl pyrrolidone K-30 and β-mercaptoethanol concentration were standardized for the procedure of DNA isolation. The quality of DNA was examined over 0.8% agarose gel using lambda uncut marker (Fermentas, Lithuania) and quantified by NanoDrop 2000 (Thermo Scientific, USA).

### Amplification, validation, and polymorphic potential evaluation

Thirty-eight oat germplasms (S1 Table) were genotyped by 22 inter simple sequence repeat (ISSR) markers (Table 1) and a DNA fingerprint was developed (S3 Table). For all the PCR amplification (Fig 1), efficiency with 22 ISSR markers was analyzed in 25μL reaction volume with 20 ng of DNA template. The PCR program was initiated as described [17]. The amplified PCR products were separated on 1.5% agarose gel (Fig 1), and amplicon size was estimated based on a 100 bp DNA ladder (Genedirex) as a reference.

### Molecular data analysis

The ISSR markers amplification profiles were scored based on the presence (1) or absence (0) of bands across all individuals (S3 Table). The amplified products were categorized as monomorphic, polymorphic, and null alleles based on amplified bands. For all the oat germplasms, Unweighted Pair Group Method with Arithmetic Mean (UPGMA) dendrogram (Fig 2) and Principal Component Analysis (PCA) (Fig 3) based on Jaccard's coefficient using DARwin6 (ver.) software [18] was constructed. The genetic structure of thirty-eight individuals of oat (Fig 4) was inferred using Bayesian algorithm-based STRUCTURE software (ver. 2.3.3) [19, 20].

## Results and discussion

### Analysis of morphological and genetic diversity

In the present study, 38 oat germplasms were taken for observation and recording of morphological data *viz.* days to 50% flowering, plant height, tillers per plant, leaf per plant, leaf length, leaf width, fresh weight (for fodder purpose), leaf area (cm$^2$), leaf area index, dry weight, harvest index, and grain yield. Among the recorded observed phenotypic traits, the grain yield and fresh weight have been taken for the selection of best-performing germplasms for food and feed purposes respectively. The statistical analysis was performed using phenotypic data to

**Table 1. List of primers and their amplification characteristics.**

| S. No. | Locus name | Primer sequence | Annealing temperature (*Tm*) ˚C | No. of alleles | Approximate size range (bp) |
|---|---|---|---|---|---|
| 1 | UBC 807 | AGAGAGAGAGAGAGAGT | 50 | 6 | 350–1700 |
| 2 | UBC 808 | AGAGAGAGAGAGAGAGG | 50 | 9 | 400–1200 |
| 3 | UBC 809 | AGAGAGAGAGAGAGAGG | 50 | 6 | 400–1100 |
| 4 | UBC 810 | GAGAGAGAGAGAGAGAT | 50 | 6 | 650–1250 |
| 5 | UBC 811 | GAGAGAGAGAGAGAGAC | 50 | 6 | 650–1300 |
| 6 | UBC 812 | GAGAGAGAGAGAGAGAA | 50 | 8 | 500–1200 |
| 7 | UBC 815 | CTCTCTCTCTCTCTCTG | 50 | 7 | 450–1400 |
| 8 | UBC 817 | CACACACACACACACAA | 50 | 7 | 700–1600 |
| 9 | UBC 818 | CACACACACACACACAG | 50 | 7 | 700–1200 |
| 10 | UBC 819 | GTGTGTGTGTGTGTGTA | 50 | 3 | 900–1300 |
| 11 | UBC 820 | GTGTGTGTGTGTGTGTT | 50 | 8 | 600–1250 |
| 12 | UBC 851 | TCTCTCTCTCTCTCTCA | 50 | 11 | 500–1600 |
| 13 | UBC 824 | TCTCTCTCTCTCTCTCG | 50 | 12 | 400–1250 |
| 14 | UBC 825 | ACACACACACACACACT | 50 | 9 | 700–1250 |
| 15 | UBC 826 | ACACACACACACACACC | 50 | 5 | 500–1300 |
| 16 | UBC 827 | ACACACACACACACACG | 50 | 8 | 200–1200 |
| 17 | UBC 834 | AGAGAGAGAGAGAGAGYT | 50 | 9 | 300–1250 |
| 18 | UBC 835 | AGAGAGAGAGAGAGAGYC | 50 | 6 | 400–900 |
| 19 | UBC 840 | GAGAGAGAGAGAGAGAYT | 50 | 7 | 550–1900 |
| 20 | UBC 841 | GAGAGAGAGAGAGAGAYC | 50 | 8 | 500–1500 |
| 21 | UBC 842 | GAGAGAGAGAGAGAGAYG | 50 | 5 | 400–1200 |
| 22 | UBC 856 | ACACACACACACACACYA | 50 | 5 | 500–1200 |

classify the oat germplasms (S2 Table). The highest grain weight was observed in OS-6 (27.33 grams) and the highest fresh weight was recorded in OL-14 (197.33). The detailed analysis of phenotypic data is available in S2 Table which can be referred to for data analysis.

In the present study 22 ISSRs have been used on 38 oat germplasms to see the genetic diversity among them. The UPGMA dendrogram (Fig 2) was constructed based on DNA fingerprinting. The relatedness between germplasms could be seen in the dendrogram and superior performing germplasm in terms of grain yield and fresh leaf weight could be selected. In Fig 2, the germplasms were clustered into two major groups i.e., I and II. Group, I have been divided into two subgroups IA and IB, further IA is subdivided into IAa and IAb. The number of alleles amplified ranged from 6–11 with size from 350-1900bp. Based on the allelic range, principal component analysis (PCA) was performed and the germplasms were grouped into 4 clusters (Fig 3). Further, the structure (Fig 4) was constructed based on genotypic data to see the relatedness among studied germplasms. Here, in Fig 4, 23 germplasms (9, 10, 13, 14, 8, 17, 7, 11, 12, 15, 16, 21, 22, 20, 23, 4, 6, 18, 19, 5, 3, 2, and 1; relative numbers represent the respective germplasm, S1 Table) out of 38 oat germplasms forms the first group, rest of the germplasms falls under the second group. The highest number of alleles was amplified by UBC 824 which can be considered the best marker among all studied ISSRs.

Bio-prospecting within oat germplasms by linking the molecular markers with desirable traits is of very potential importance [21]. The variability among oat germplasms could be harnessed to enhance crop productivity. Therefore, the molecular markers like ISSR markers could be used to exploit the genetic diversity of oats and selected desirable trait containing germplasms can be used for the marker-assisted breeding program. With this analysis i.e., dendrogram, PCA, and structure analysis, it is evident that the studied oat germplasms could be

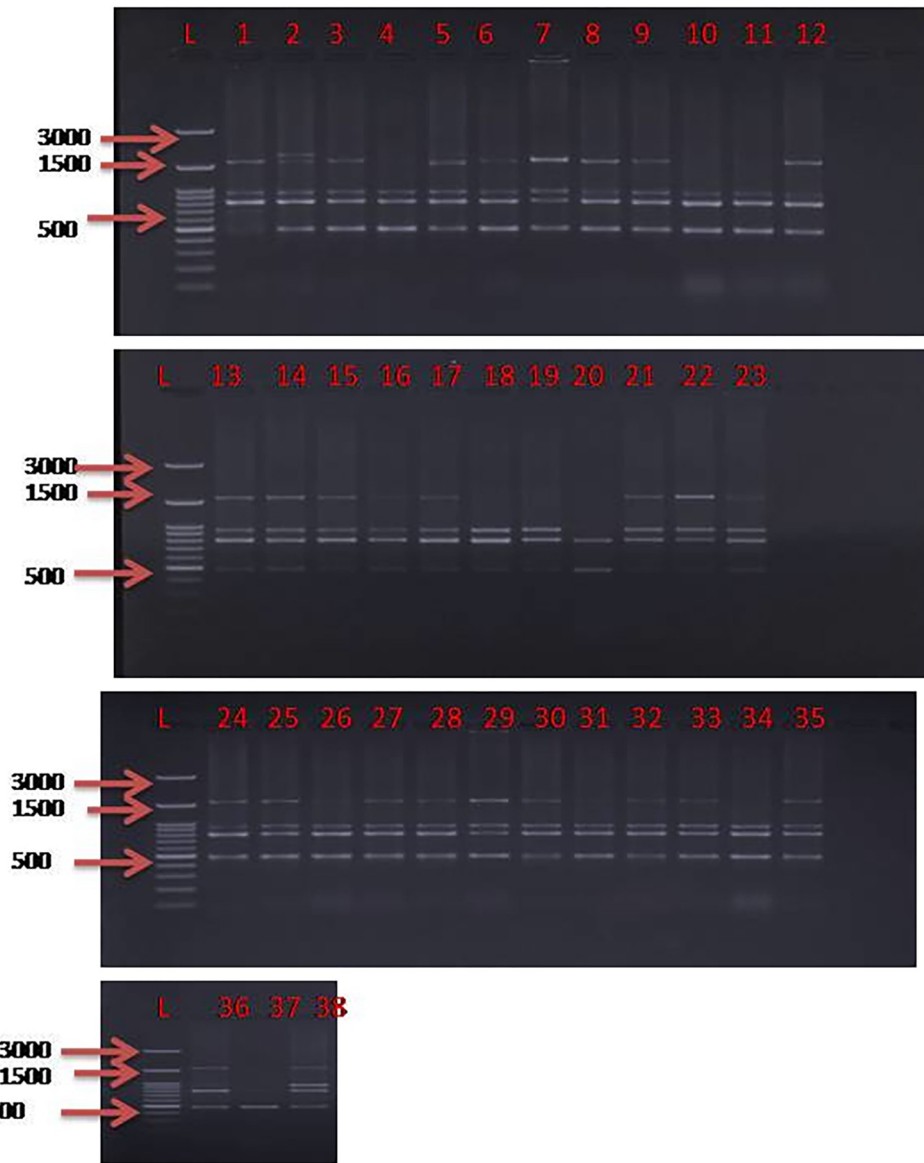

**Fig 1. PCR amplification profile of 38 oat germplasms using UBC 826 Inter Simple Sequence Repeats (ISSRs) markers along with ladder.**

grouped based on genotyping using ISSR markers. The grain yield and fresh weight observed in the taken oat germplasm were influenced by both genotypic content and environment. The variable gradient of soil micronutrients especially potassium and copper play a very significant role in the oat and wheat grain yield [22, 23]. But many times, the nutrient gradient may also give false positive results. To avoid the influence of false positive results due to environmental effects, genotypic analysis was conducted to select the best-performing germplasm. Further, both the data were taken together to select the best germplasm. The relatedness among organisms and selection of desirable traits containing oat germplasms by phenotyping and genotyping could be used for crop improvement through marker-assisted selection.

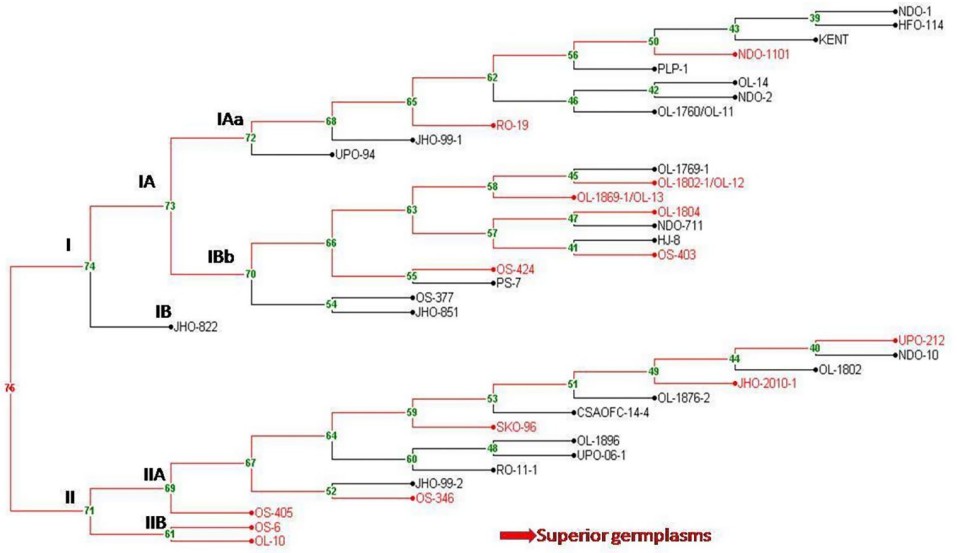

**Fig 2. Dendrogram of 38 *Avena sativa L*. based on 22 Inter Simple Sequence Repeats (ISSRs) markers.**

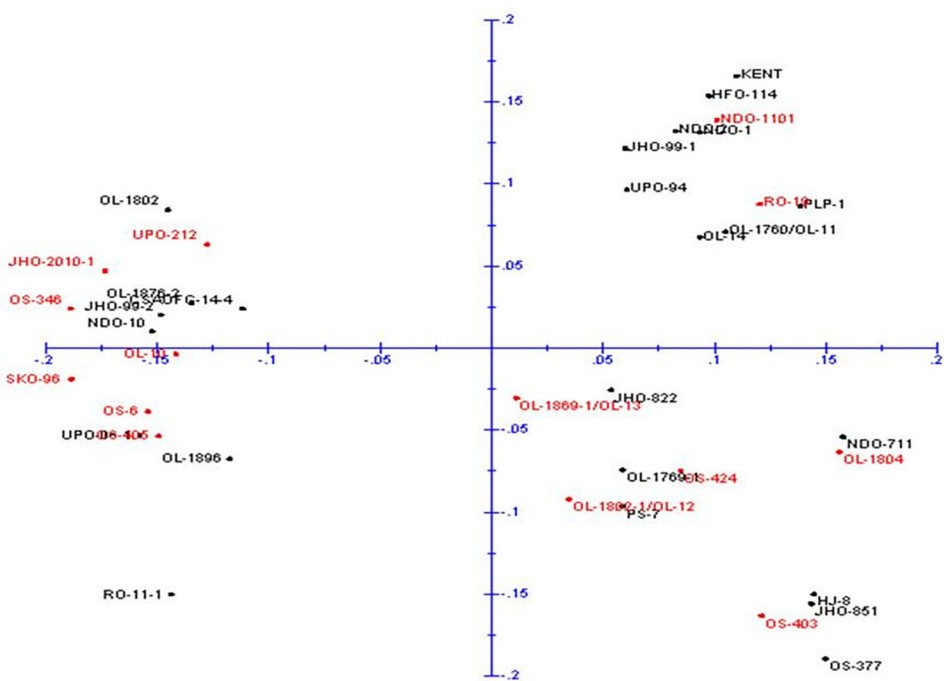

**Fig 3. Inter Simple Sequence Repeats (SSR) marker-based principal component analysis showing two-dimensional distributions of oat germplasms.**

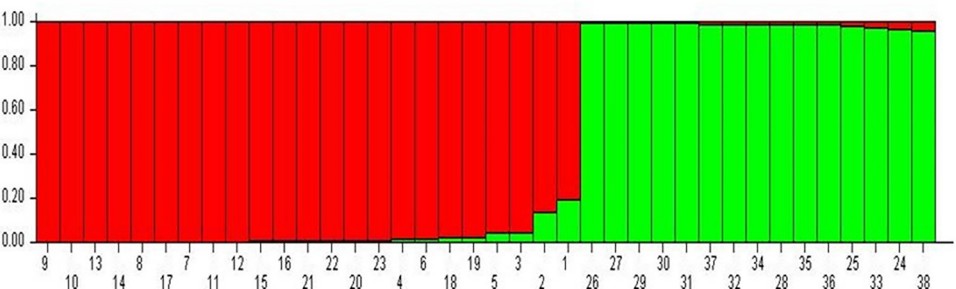

**Fig 4. Bayesian model-based genetic clustering of 38 oat germplasms.**

## Classification of high grain yield (for food purposes), high fresh weight (for fodder purposes) low grain yield, and low fresh weight based on phenotypic data

Among several agriculturally important traits concerned with food and feed, grain yield and fresh weight of oats are very crucial. Therefore, the present study was conducted to identify high grain yield and high fresh weight containing germplasms by correlating the phenotypic data with DNA fingerprint-based dendrogram (Fig 2), principal component analysis (Fig 3), and phenotypic data based heat map (S1A, S1B and S2A, S2B Figs). After careful analysis of phenotypic and genotypic data, we have identified eight oat germplasms (OS-424, OS-403, NDO-1101, OL-10, UPO-212, OS-405, OS-6, and OS-346) having high grain yield ranging from 16gm to 27.3 gm per plant (S1 Table, Fig 1A). Among these eight high grain yield germplasms, all showed a strong correlation with the eight individuals but they were negatively correlated with UPO-94 (low grain yield) (Fig 1B). The correlation heatmap matrix of the high-grain yield germplasms was further confirmed by a fingerprint-based dendrogram (Fig 2). Further, eight germplasms were identified as superior under high fresh weight (for fodder purposes) NDO-711, RO-19, OL-14, OL-1760/OL-11, NDO-10, UPO-212, UPO-06-1, and RO-11-1 ranging from 166gm to 197.33 gm per plant (S1 Table, S2A Fig). These eight high fresh weights germplasms, all showed a strong correlation with the eight individuals but they were negatively correlated with UPO-94 (low fresh weight) (S2b Fig). Further UPO-94 showed negative correlation with NDO-711, RO-19, OL-14, OL-1760/OL-11, NDO-11-1 germplasms. The correlation heatmap matrix (S2B Fig) of the high fresh weight germplasms was further confirmed by DNA fingerprint-based dendrogram (Fig 2) and heatmap data (S2A Fig). The phenotypic data taken on the open field depends on the various biotic and abiotic factors. These factors influence the observation values that ultimately affect the accuracy of the data. Here, the classification is based on morphological data that is highly influenced by environmental conditions and may be changed in the coming season. Hence, classification needs genotypic data based on DNA fingerprints to make sure that the phenotypic data is not environmentally biased.

## Classification of high grain yield (for food purposes), high fresh weight (for fodder purposes) low grain yield, and low fresh weight based on genotypic data

The genotypic data collected based on the scoring of amplified PCR products were used to see the relatedness among the selected genotypes. Studies of genetic diversity enhance our precision to select the cross-compatible closely related but diverse in nature germplasms to improve

overall crop productivity [24]. To classify the studied germplasm on the basis of genotypic data derived from molecular markers, the UPGMA dendrogram (Fig 2) was constructed the divides the total germplasms into two main clusters (cluster I and II) and two sub-clusters (IA and IB). Further, the subcluster IA was divided into IAa and IAb. The sub-cluster IA had 22 oat germplasms. The IAa and IAb contain high-yield germplasms along with low grain-yielding germplasms. Most of the germplasms of IAa and IAb including UPO-94 (low grain yield-16gm) have low grain yield germplasms except RO-19 and NDO-1101 of group IAa and OL-1802-1/ OL-12, OL-1869-1/OL-13, OL-1804, OS-403, and OS-424 of group IAb (Fig 2). The cluster IB represented only one genotype JHO-822 and was characterized under low grain yielding germplasms. Further, the analysis discriminated the sub-cluster II into IIA and IIB. Cluster IIA represented thirteen germplasms. Interestingly, most of the germplasms of the cluster IIA showed high grain yield including UPO-212, JHO-2010-1, SKO-96, OS-346and OS-405 except OL-1802, NDO-10, OL-1876-2, CSAOFC-14-4, OL-1896, UPO-06-1, RO-11-1, and JHO-99-2 (Fig 2). The cluster IIB represented two germplasms OS-6 and OL-10 that were characterized by high grain yield. After correlating the morphological and genotypic data, we selected the high grain yield and high fresh weight oat germplasms. Further, the high trait contrasting germplasm were selected OS-6 (high grain yield, cluster-IIB) and UPO-94 (low grain yield, cluster- IAa), and OL-14 (Cluster-IAa, high fresh weight for fodder purpose) and UPO-94 (low fresh weight for fodder purpose, cluster- IAa) were selected based on DNA genotypic data.

The correlation of DNA fingerprint (S3 Table), heat map analysis (S1A and S2A Figs), and heatmap matrix (S1B and S2B Figs) revealed that the high grain yield germplasms OS-424, OS-403, NDO-1101, OL-10, UPO-212, OS-405, OS-6, and OS-346 and fresh weight germplasms NDO-711, RO-19, OL-14, OL-1760/OL-11, NDO-10, UPO-212, UPO-06-1, and RO-11-1germplasms were best germplasms which had more than 16-gram grain yield and 166.33-gram fresh weight compared to the low grain yielding and low fresh weight UPO-94 (9.56 and 65.33 gram respectively) germplasm.

Transformation and plotting of the abundance data using principal component analysis (PCA, Fig 3) separated the samples which reflected the variation of oat germplasms possibly due to photoperiod sensitivity and vernalization also suggested in other studies [25, 26]. Moreover, the PCA (Fig 3) and DNA fingerprint (Fig 1) analysis characterized the 38 oats genotypes into two sections. Germplasms under section I was categorized as high grain yield and high fresh weight while section II was under low grain yield and low fresh weight (Fig 3). The results of PCA followed by the confirmation of heat map (S1A and S2A Figs) and correlation heatmap matrix (S2A and S2B Fig) analyses revealed that OS-424, OS-403, NDO-1101, OL-10, UPO-212, OS-405, OS-6, and OS-346 were high grain yielding genotypes, and NDO-711, RO-19, OL-14, OL-1760/OL-11, NDO-10, UPO-212, UPO-06-1, and RO-11-1 displayed high fresh weight under section I (Fig 3). Further, low grain yield and low fresh weight yielding germplasms were grouped in section II (Fig 3). The germplasm UPO-94 was characterized under section II (Fig 3). It could be noted by the correlation of DNA genotype, PCA, heatmap, and correlation heatmap matrix that OS-424, OS-403, NDO-1101, OL-10, UPO-212, OS-405, OS-6, and OS-346 showed high grain yield, and NDO-711, RO-19, OL-14, OL-1760/OL-11, NDO-10, UPO-212, UPO-06-1, and RO-11-1genotypes showed high fresh weight as compared to UPO-94. Keeping these oat germplasms having desirable traits could be used for trait transfer via conventional breeding/marker-assisted backcross breeding programs [27]. Further, the selected germplasms will be analyzed in the future for the high palatability of grains and leaves.

After having a serious look at the available scientific literature on various Gramineae family crops [28–30], the present study could be the first and novel report involving the ISSR markers for the characterization of high grain and fresh-weight yielding genotypes in oats. A set of 22

ISSRs have been used to expedite the molecular breeding of oats. The high cross-transferability and polymorphism of these ISSR markers further reveal their novelty [31]. Utilization of these novel ISSRs markers in diversity analysis and population structure characterization of 38 oat germplasms suggest their wider efficacy on a superior scale for molecular breeding studies in oats. Based on the DNA fingerprint analysis, the study revealed that 38 oats germplasms were highly diverse. Amongst these 38 germplasms; eight (OS-424, OS-403, NDO-1101, OL-10, UPO-212, OS-405, OS-6, and OS-346) were identified superior under high grain yield ranging from 16 gm to 27.3 whereas the other eight (NDO-711, RO-19, OL-14, OL-1760/OL-11, NDO-10, UPO-212, UPO-06-1, and RO-11-1) germplasms were identified superior under high fresh weight (for fodder purpose) ranging from 166 gm to 197.33 gm. These identified germplasms would act as breeding material for the introgression of high fresh weight and high grain yield traits into elite cultivars.

## Supporting information

**S1 Fig. a- Heat map depicting grain yield profile of 38 oat germplasms, b-Correlation heat-map matrix of grain yield profile of 38 oat germplasms.**
(TIF)

**S2 Fig. a-Heat map depicting fresh weight (for fodder purpose) profile of 38 oat germ-plasms, b-Correlation heatmap matrix of fresh weight (for fodder purpose) profile of 38 oat germplasms.**
(TIF)

**S1 Table. Description of the 38 germplasms used for the identification of high grain yield-ing and fresh weight (for fodder purpose) germplasms of oats.**
(DOCX)

**S2 Table. Performance of oat genotypes for 12 characters along with phenotypic and geno-typic data analysis using statistical tools.**
(DOCX)

**S3 Table. DNA fingerprint of 38 oats with 22 ISSR markers.**
(XLSX)

**S1 Raw images.**
(PDF)

## Author Contributions

**Conceptualization:** Deepanshu Jayaswal, Kuldip Jayaswall, Sanjay Kumar.

**Data curation:** Raj Kumar, Deepanshu Jayaswal, Kuldip Jayaswall, Arvind Nath Singh.

**Formal analysis:** Raj Kumar, Smriti Varghese, Deepanshu Jayaswal, Kuldip Jayaswall, Gaurav Mishra, Sanjay Kumar.

**Investigation:** Deepanshu Jayaswal, H. G. Prakash.

**Methodology:** Deepanshu Jayaswal, Kuldip Jayaswall.

**Resources:** R. P. Vyas, H. C. Singh, Arvind Nath Singh.

**Software:** Deepanshu Jayaswal, Gaurav Mishra.

**Supervision:** Deepanshu Jayaswal, Kuldip Jayaswall, R. P. Vyas.

**Validation:** Kuldeep Yadav.

**Visualization:** Kuldeep Yadav.

**Writing – original draft:** Raj Kumar.

**Writing – review & editing:** Smriti Varghese, Deepanshu Jayaswal, Kuldip Jayaswall, H. C. Singh, H. G. Prakash, Arvind Nath Singh.

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
