## [Decision Letter · Decision Letter 0]

23 Nov 2022

PONE-D-22-26734

Agro-morphological variability, population structure, and genetic diversity in oat germplasms with special emphasis on food and feed

PLOS ONE

Dear Dr. Jayaswal,

Thank you for submitting your manuscript to PLOS ONE. After careful consideration, we feel that it has merit but does not fully meet PLOS ONE’s publication criteria as it currently stands. Therefore, we invite you to submit a revised version of the manuscript that addresses the points raised during the review process.

We look forward to receiving your revised manuscript.

Kind regards,

Tzen-Yuh Chiang

Academic Editor

PLOS ONE

Journal Requirements:

Reviewers' comments:

Reviewer's Responses to Questions

**Comments to the Author**

1. Is the manuscript technically sound, and do the data support the conclusions?

Reviewer #1: Yes

2. Has the statistical analysis been performed appropriately and rigorously? 

Reviewer #1: Yes

3. Have the authors made all data underlying the findings in their manuscript fully available?

Reviewer #1: Yes

4. Is the manuscript presented in an intelligible fashion and written in standard English?

Reviewer #1: Yes

5. Review Comments to the Author

Reviewer #1: Dear Editor

I have gone through paper entitled “Agro-morphological variability, population structure, and genetic diversity in oat germplasm with special emphasis on food and feed”.

Although, the several studies have been published on the same topic, nothing new in this paper. However, the paper is generally well organized, and the introduction gave a literature review of related topics, and summarized the methods proposed in this paper. In order to make the paper easy to understand, appropriate figures and tables were also given. Therefore, I think that the paper could be published in the Journal with some modifications.

Here my comments and suggestions:

• Still some grammatical errors are present, that should be removed before publication.

• Title should be more appropriate

• Key words are not appropriate

• Write full forms of all abbreviations at least once.

• Sampling and data recording is clear and significant.

• All Figures are poor and not clear, should be replace with clear and readable images.

• Also check all references in the text and reference list carefully some are missing from the list or not cited in the text.

• Most importantly, the discussion section needs to be strengthened to highlight the importance of the work.

For further justification of the study, kindly incorporate the below current review, please.

Al-Yasari MNH (2022). Potassium and nano-copper fertilization effects on morphological and production traits of oat (Avena sativa L.). SABRAO J. Breed. Genet. 54(3): 678-685. http://doi.org/10.54910/sabrao2022.54.3.20.

Abdel-Lateif KS, Hewedy OA (2018). Genetic diversity among Egyptian wheat cultivars using SCoT and ISSR markers. SABRAO J. Breed. Genet. 50: 36–45.

Qulmamatova DE, Baboev SK, Buronov AK (2022). Genetic variability and inheritance pattern of yield components through diallel analysis in spring wheat. SABRAO J. Breed. Genet. 54(1): 21-29. http://doi.org/10.54910/sabrao2022.54.1.3

Helsel DG, Skrdla RK (1983). Breeding for grain yield in oats (Avena sativa L.). SABRAO J. Breed. Genet. 15(2): 139-145.

Swailam MA, Mowafy SAE, El-Naggar NZA, Mansour E (2021). Agronomic responses of diverse bread wheat genotypes to phosphorus levels and nitrogen forms in a semiarid environment. SABRAO J. Breed. Genet. 53: 592-608. https://doi.org/10.54910/sabrao2021.53.4.4.

6. PLOS authors have the option to publish the peer review history of their article (what does this mean?). If published, this will include your full peer review and any attached files.

Reviewer #1: **Yes: **PROF. DR. NAQIB Ullah KHAN

---

## [Author Response · Author response to Decision Letter 0]

23 Dec 2022

To,

The Editor

PLOS ONE

Please find enclosed our detailed responses to the reviewers’ and editor’s comments for the manuscript “Agro-morphological and genetic variability analysis in oat germplasms with special emphasis on food and feed”. The Editor’s and reviewer’s comments were very constructive and mainly concerned (1) Format, title and pictures quality MS, (2) References and the grammar. We appreciate the time and effort that you and the reviewers dedicated to providing feedback on our manuscript. We have dealt with all of the critical comments in full, and revised the manuscript. We hope that you agree that we have satisfactorily dealt with the reviewer’s comments in full.

Sincerely,

Deepanshu Jayaswal on behalf of all authors.

Response to Editor

Comment 1: A rebuttal letter that responds to each point rose by the academic editor and reviewer(s). You should upload this letter as a separate file labeled 'Response to Reviewers'.

Response: The Response to Reviewers file has been attached.

Comment 2: A marked-up copy of your manuscript that highlights changes made to the original version. You should upload this as a separate file labeled 'Revised Manuscript with Track Changes'.

Response: 'Revised Manuscript with Track Changes' with suggested comments has been attached.

Comment 3: An unmarked version of your revised paper without tracked changes. You should upload this as a separate file labeled 'Manuscript'.

Response: Submitted as suggested.

 Response to Journal Requirements

Comment 1: Please ensure that your manuscript meets PLOS ONE's style requirements, including those for file naming.

Response: The whole manuscript has been considered as per the journal requirement and completed as per requirement.

Comment 2: In your Data Availability statement, you have not specified where the minimal data set underlying the results described in your manuscript can be found. PLOS defines a study's minimal data set as the underlying data used to reach the conclusions drawn in the manuscript and any additional data required to replicate the reported study findings in their entirety. All PLOS journals require that the minimal data set be made fully available. 

Response: The minimal data requirements have been provided in all the attached supplementary tables.

Comment 3: PLOS ONE now requires that authors provide the original uncropped and unadjusted images underlying all blot or gel results reported in a submission’s figures or Supporting Information files. 

Response: We would like to ask for an apology for this. The student has taken the gel images and cropped without saving the original images. Therefore, we have all the images but in cropped version. For this we are in regret and can not produce the uncropped image. Meantime we will try to recover the images from the gel doc system where the images were taken.

Comment 4: Please include captions for your Supporting Information files at the end of your manuscript, and update any in-text citations to match accordingly.

Response: Legends captions for all the figures and tables are given after references.

Comment 5: Please review your reference list to ensure that it is complete and correct. If you have cited papers that have been retracted, please include the rationale for doing so in the manuscript text, or remove these references and replace them with relevant current references. Any changes to the reference list should be mentioned in the rebuttal letter that accompanies your revised manuscript. If you need to cite a retracted article, indicate the article’s retracted status in the References list and also include a citation and full reference for the retraction notice.

Response: We have gone through the MS and checked for all references.

Response to Reviewers

Comment 1: Still some grammatical errors are present, that should be removed before publication.

Response: The authentic purchased software was used to rectify the grammatical mistekes.

Comment 1: Title should be more appropriate

Response: As the term diversity itself includes population structure analysis therefore, title has been modified to make it bit short. Agro-morphological and genetic variability analysis in oat germplasms with special emphasis on food and feed

Comment 2: Key words are not appropriate

Response: Keywords has been replaced by those which are involved many times in the MS and represent the research work.

Comment 3: Write full forms of all abbreviations at least once.

Response: Please accept our apology for such small but key mistakes. Now, we have corrected as per suggestion.

Comment 4: Sampling and data recording is clear and significant.

Response: We are highly grateful for appreciation.

Comment 5: All Figures are poor and not clear, should be replacing with clear and readable images.

Response: The entire pictures dpi has been upgraded up to 600 dpi to enhance the visibility.

Comment 6: Also check all references in the text and reference list carefully some are missing from the list or not cited in the text.

Response: We have carefully checked the total number of references given and their citations in the MS text and rectified as per necessity.

Comment 7: Most importantly, the discussion section needs to be strengthened to highlight the importance of the work.

Response: Discussion of each finding has been explained where needed as per suggestion.

---

## [Decision Letter · Decision Letter 1]

3 Jan 2023

Agro-morphological and genetic variability analysis in oat germplasms with special emphasis on food and feed

PONE-D-22-26734R1

Dear Dr. Jayaswal,

We’re pleased to inform you that your manuscript has been judged scientifically suitable for publication and will be formally accepted for publication once it meets all outstanding technical requirements.

Kind regards,

Tzen-Yuh Chiang

Academic Editor

PLOS ONE

Additional Editor Comments (optional):

Reviewers' comments:

Reviewer's Responses to Questions

**Comments to the Author**

1. If the authors have adequately addressed your comments raised in a previous round of review and you feel that this manuscript is now acceptable for publication, you may indicate that here to bypass the “Comments to the Author” section, enter your conflict of interest statement in the “Confidential to Editor” section, and submit your "Accept" recommendation.

Reviewer #1: All comments have been addressed

2. Is the manuscript technically sound, and do the data support the conclusions?

Reviewer #1: Yes

3. Has the statistical analysis been performed appropriately and rigorously? 

Reviewer #1: Yes

4. Have the authors made all data underlying the findings in their manuscript fully available?

Reviewer #1: Yes

5. Is the manuscript presented in an intelligible fashion and written in standard English?

Reviewer #1: Yes

6. Review Comments to the Author

Reviewer #1: Yes, and as per previous review comments, the the article is well improved, and it can be published now, please,.

7. PLOS authors have the option to publish the peer review history of their article (what does this mean?). If published, this will include your full peer review and any attached files.

Reviewer #1: **Yes: **PROF. DR. NAQIB Ullah KHAN

---

## [Editor Report · Acceptance letter]

27 Jan 2023

PONE-D-22-26734R1 

Agro-morphological and genetic variability analysis in oat germplasms with special emphasis on food and feed 

Dear Dr. Jayaswal:

I'm pleased to inform you that your manuscript has been deemed suitable for publication in PLOS ONE. Congratulations! Your manuscript is now with our production department. 

Kind regards, 

on behalf of

Dr. Tzen-Yuh Chiang 

Academic Editor

PLOS ONE